# Study on Influencing Factors of Ground Pressure Behavior in Roadway-Concentrated Areas under Super-Thick Nappe

**DOI:** 10.3390/ma16010089

**Published:** 2022-12-22

**Authors:** Ruojun Zhu, Xizhan Yue, Xuesheng Liu, Zhihan Shi, Xuebin Li

**Affiliations:** 1College of Mining, China University of Mining and Technology, Xuzhou 221116, China; 2China Coal Xinji Energy Co., Ltd., Huainan 232001, China; 3College of Energy and Mining Engineering, Shandong University of Science and Technology, Qingdao 266590, China

**Keywords:** super-thick nappe, roadway-concentrated areas, influencing factors, ground pressure behavior, numerical simulation

## Abstract

During the mining activity under the super-thick nappe formed by thrust fault, the law of mine pressure behavior is complex, and it is difficult to control the deformation and failure of surrounding rock. Combined with the actual engineering conditions, the influence of different roof lithology conditions, the thickness of nappe, the mining height, the size of the barrier coal pillar, and the creep time on mine pressure behavior was studied by UDEC numerical simulation software. The results showed that with the advancement of the coal face, due to the influence of the mining of the coal face and the slip dislocation of the super-thick nappe along the thrust faults, the roof-to-floor convergence, the two-sided convergence, and the maximum concentrated stress in the roadway-concentrated areas are significantly increased. For the above five influencing factors, the greater the thickness of the nappe and the mining height, the longer the creep time, and the stronger the ground pressure behavior. The larger the size of the barrier coal pillar, the stronger the roof lithology, and the gentler the ground pressure behavior. The research results can provide some reference for monitoring the law of ground pressure behavior in roadway-concentrated areas under super-thick nappe.

## 1. Introduction

The exploitation of coal resources under complex geological conditions has always been a matter of great concern in the coal industry [1,2,3,4]. Thrust fault is a special type of geological structure. It is a large thrust fault with a dip angle of about 30° or less [5,6,7]. Such faults tend to cause their hanging walls to move thousands or even tens of thousands of meters over long distances along the fault plane, creating a massive rock mass structure known as a nappe [8]. The structure of the nappe will lead to the complex law of ground pressure behavior and unclear influencing factors, which will seriously affect the safety and stability of the roadway.

In recent years, scholars at home and abroad have conducted a lot of research on the law of ground pressure behavior and its influencing factors under the influence of faults, such as the use of FLAC 3D and boundary element numerical simulation methods. Batugin et al. [9,10,11,12,13,14] obtained the best supporting parameters of surrounding rock under fault conditions and the influence of different types of faults on ground pressure behavior. Sainoki et al. [15,16,17,18,19,20,21,22,23,24] obtained the relationship between mining activities and microseismical phenomena and the influence of different mining directions on the law of ground pressure behavior in coal face. Islam et al. [25], taking Barapukuria coal mine in Bangladesh as an example, found that the deformation and stress field of the fault and its surrounding rock change significantly under mining disturbance, and stress concentration occurs at the end of the fault, using a model test or a similar material test method. Zhang et al. [26,27,28,29,30,31] studied the overburden strata movement law, mine pressure characteristics, dynamic response, slip precursor information, and instability transient process characteristics before and after the activation of a thrust fault, and the related factors affecting the law of ground pressure behavior were obtained. Combined with elastic-plastic mechanics theory, Han et al. [32,33,34,35] studied the stress distribution and failure evolution process of surrounding rock under different conditions. Using the field monitoring method, Ji et al. [36,37] studied the influence of the relative position of the coal face and the fault on the law of mine pressure behavior, and the fault activation process is inversed by the microseismical data of the mine. Using a method based on a probabilistic approach to assessing the rock strength, Begalinov et al. [38] obtained the proportion of mining active fault instability under the influence of faults by studying the physical and mechanical properties of faults, which provides a basis for determining the means of support. Apart from that, Wang et al. [39] proposed a mechanical model considering the stress redistribution caused by coal face mining, calculating the stress distribution of the fracture surface and explaining the characteristics of the fault slip. Dokht et al. [40] concluded that mining activities of the mine cause the redistribution of roadway stress, which leads to the activation of faults and eventually causes small unnatural earthquakes. The above research has played an important role in the study of the law of ground pressure behavior and its influencing factors under the influence of faults and has effectively promoted the process of quantitative research on the control of roadway-surrounding rock under the influence of complex geological conditions.

However, due to the particularity of the thrust fault, the influence of the super-thick nappe on the ground pressure behavior in the concentrated area of the lower roadway is not clear, and the influencing factors are rarely studied. Therefore, this paper takes the mining of No. 360801 coal face under the super-thick nappe structure of Xinji No. 1 Mine as the engineering background. Firstly, the main factors affecting the ground pressure behavior in the roadway-concentrated areas under the super-thick nappe are determined. Then, a numerical simulation model under different geological and mining conditions is established by using the Universal Distinct Element Code (UDEC 6.0 v6.0.336) discrete element numerical simulation software, and then the law of ground pressure behavior in roadway-concentrated areas under super-thick nappe is explored. The research results can provide some reference for monitoring the law of ground pressure behavior in roadway-concentrated areas under super-thick nappe. The remainder of this manuscript is arranged as follows. Section 2 contains the location and geological conditions of the No. 360801 coal face, and then several important factors affecting the law of mine pressure behavior are analyzed. In Section 3, the numerical simulation model is established for different influencing factors. In Section 4, the specific influence of different factors on the mine pressure behavior is analyzed according to the simulation results. In Section 5, the numerical simulation results are summarized and analyzed, and conclusions are drawn.

## 2. Project Profile

The No. 36 district of Xinji No. 1 Mine is located in the southwest of the Xinji minefield. There is a large range of Fufeng thrust faults above the district. The overall trend of the fault is southward, and the dip angle changes greatly along the dip direction. The strike zone is 30°–80°, and the middle dip angle gradually slows down, generally to about 5°. The local section is nearly horizontal or inclined north, and the fault drop gradually decreases from south to north. The fault zone is undulating in the strike and dips with a width of 0~27.25 m. It is composed of broken brecciated limestone, mudstone, and carbonaceous mudstone. It is easy to loosen and break, and the local cementation is good. It is a typical thrust fault. Due to the influence of faults, there is a nappe structure with a thickness of 351~562 m above the district (Figure 1). The structure of the nappe is mainly composed of gneiss. The lithology is mainly granite gneiss and hornblende gneiss, showing gray mixed with a gray-black, brown, fresh surface, a scale granular crystal structure, and a gneiss structure. The No. 360801 coal face is located in the north of the west wing of the No. 36 district, the main 8# coal-mining area. The outer section of the No. 360802 machine roadway and the outer section of the No. 360803 track crossheading are in the east of the coal face, the design cut position near the auxiliary No. 13 exploration line is in the west, the track crossheading of the No. 360803 coal face is in the south, and the F10 protective coal pillar line is in the north. The layout of the No. 360801 coal face is shown in Figure 2. The 8# coal is mainly bright coal with a thickness of 2.6~3.6 m and an average thickness of 3.1 m. The immediate roof is mudstone, and the rock stability is general. The basic roof is fine sandstone with a maximum thickness of 13.5 m, and the rock layer is stable. The floor is mudstone, in a dense block with a parallel bedding development, with a thickness of 5.54~9.15 m. The specific rock-layer histogram is shown in Figure 3.

Since the thickness of the nappe is too large, the mine pressure behavior of the coal face cannot be accurately measured, and the specific influence of the change of mining conditions and geological conditions on the mine pressure cannot be confirmed. In the actual engineering survey, we found that the mine pressure behavior in many areas of the roadway-concentrated area is very strong. The −706-rail crosscuts and transport rise-entry show serious deformation and failure of the surrounding rock, poor stability, and other issues, as shown in the Figure 4a,b. Often, multiple repairs are needed to ensure normal use, seriously affecting the safety of the mine production. Therefore, it is necessary to study the influencing factors of mine pressure behavior.

## 3. Factors and Methods

### 3.1. Analysis of the Main Factors 

#### 3.1.1. Geologic Factor

(1)Roof lithology

Roof lithology refers to the mechanical properties of the rock layer on the top of the roadway in a mine. When the roof lithology is different, the stability of roof strata is also different [41,42,43]. The deformation of the roof and the degree of rock fragmentation are quite different, which leads to the possibility of rock failure under the long-term action of a small external load or a small load. The direct roof of the 8# coal in the No. 36 district of the Xinji No. 1 Mine is mainly mudstone and sandy mudstone. The stability of rock strata is general, and the stability of rock strata is poor in the structural development area. The direct roof is scoured by the main roof of sandstone, which leads to direct contact between the main roof sandstone and the coal seam. At the same time, it is affected by the structure of the super-thick nappe, which makes the ground pressure behavior law more complicated, and then affects the safety and stability of the roadway-concentrated areas under the super-thick nappe.

(2)Thickness of nappe

Nappe is a huge rock mass structure formed by thrust faults that cause the hanging wall to move thousands or even tens of thousands of meters along the fault plane. The nappe structure will lead to inaccurate identification of the coal seam buried depth and distribution, and the inability to determine the structure development and distribution for the calculation and accurate measurement of the mine pressure caused great difficulties. When the thickness of the nappe greatly changes, the ground pressure behavior also dramatically changes. Therefore, it is particularly important to determine the relationship between the law of ground pressure behavior and the thickness of the nappe. Combined with the actual engineering conditions, the thickness of the overlying nappe in the No. 36 district of the Xinji No. 1 Mine is 351~562 m, and the thickness greatly varies. Therefore, the ground pressure behavior in different positions of the roadway-concentrated areas under the super-thick nappe is different.

#### 3.1.2. Mining Technological Factors

(1)Mining height

Mining height refers to the distance difference between upper repeated mining and lower normal mining for inclined coal seam mining. The mining height of the coal face is directly related to the law of mine pressure behavior, and it is also the fundamental factor of the deformation and failure of the overlying strata. In combination with the engineering practice, in the No. 36 district of the Xinji No. 1 Mine, with the progress of the 8# coal-mining work, the stress of the rock mass around the roadway is redistributed, resulting in a significant change in the law of ground pressure behavior in the roadway-concentrated areas under the super-thick nappe. If the mining height of the coal face is adjusted, the roadway stress will be redistributed, and the actual ground pressure behavior law in the roadway-concentrated areas is extremely complex.

(2)Rise-entry barrier coal pillar size

The barrier coal pillar refers to the part of the coal body that is specially left underground and is not mined, and the purpose is to protect the above-mentioned protected objects inside the rock stratum and on the surface from mining. Due to the excavation of roadways and the mining of coal seams, the stability of roadways in roadway-concentrated areas under super-thick nappe is reduced, making the district within their influence range vulnerable to damage. To protect the rise-entry of the district and maintain the stability of roadways, it is necessary to setup a barrier coal pillar. However, with the change of the size of the rise-entry barrier coal pillar, the ground pressure behavior of roadway-concentrated areas also changes, and under the combined action of it and super-thick nappe, the influence law of roadway-concentrated areas under the nappe is more complicated.

#### 3.1.3. Time Factor

The phenomenon that the deformation of rock increases with time under the continuous action of an external force whose size and direction do not change is called creep. The roof rack of the roadway is subjected to external force for a long time, and its deformation is increasing, and the stability is reduced. If its creep cannot be stabilized at a certain limit value, the breaking characteristics will be different from the creep failure of instantaneous failure, thus changing the ground pressure behavior in the roadway-concentrated areas. Therefore, the creep time also has an important influence on the ground pressure behavior in the roadway-concentrated areas under the super-thick nappe.

### 3.2. Construction of the Numerical Model

To study the influence of different factors on the ground pressure behavior in the roadway-concentrated areas under the super-thick nappe, based on the actual geological conditions of the No. 360801 coal face in the Xinji No. 1 Mine, the −706-rail crosscut, rail rise-entry, transport rise-entry, and return air rise-entry in this district are the most frequently used roadways and serve as the main transportation and return air tasks in the district. At the same time, the above four roadways are located in or through the tendency center of the No. 360801 coal face, and the research is representative. Therefore, the above four roadways were selected as the main research objects. Establishment of a numerical simulation model was performed using UDEC discrete element software [44]. Under the influence of different roof lithology conditions, the thickness of nappe, the mining height, the size of the barrier coal pillar, and the creep time, the law of ground pressure behavior in roadway-concentrated areas caused by mining was analyzed. The numerical simulation foundation model was established according to actual engineering geological conditions. The thickness of the nappe was 400 m, the mining height was 3 m, and the size of the barrier coal pillar was 190 m. The model size was length × height = 1000 m × 450 m. The boundary condition of the model was to apply the X-axis horizontal constraint on the right boundary of the model and the Y-axis vertical constraint on the bottom of the model. The vertical load was applied on the top of the model to simulate the weight of the overlying strata. The self-weight stress generated by the actual buried depth was used as the load on the top-end face. The upper self-weight stress was applied in the horizontal direction multiplied by the lateral pressure coefficient, λ, to simulate the initial ground stress. The specific numerical model is shown in Figure 5. The mechanical parameters of the whole rock stratum of the model are shown in Table 1.

To study the influence of different factors on the ground pressure behavior of roadway-surrounding rock, the numerical model under different conditions was established by adjusting the above basic model. At the same time, in the simulation process, the corresponding stress and displacement monitoring points were set at the top and bottom plates of the above four roadways and on the left side (near the side of the No. 360801 coal face) to monitor the roof-to-floor convergence, two-sided convergence, the maximum concentrated stress, and its position changes, to clarify the law of ground pressure behavior in the roadway-concentrated areas under the influence of different factors.

### 3.3. Simulation Scheme Design under Different Influence Factors

#### 3.3.1. Geologic Factor

(1)Roof lithology

To deeply analyze the influence of roof lithology on ground pressure behavior in roadway-concentrated areas under super-thick nappe, under the condition of ensuring other conditions are unchanged, the roof lithology of the No. 360801 coal face was set as very soft, soft, softer, and stiff. Different lithology is to be achieved by the assignment command in numerical simulation. The stress and displacement changes of roadway-surrounding rock after coal face mining and the influence law of roof lithology on ground pressure behavior were obtained. The specific scheme is shown in Table 2.

(2)Thickness of nappe

To ensure other conditions are unchanged, the specific influence of the thickness variation of the nappe on the ground pressure behavior of the roadway was determined. The thickness of the nappe was set to 100, 250, 400, and 550 m, respectively. By observing the changes of stress and displacement of the surrounding rock of the roadway after mining in the coal face, the influence of the different thicknesses of nappe on the ground pressure behavior was explored.

#### 3.3.2. Mining Technologic Factors

(1)Mining height

To explore the influence of mining height on the ground pressure behavior in the roadway-concentrated areas under the super-thick nappe, the other conditions were also guaranteed. The mining height was set to 1, 3, 5, and 7 m, respectively. We analyzed and compared the changes of stress and displacement of roadway-surrounding rock after coal face mining to explore the law of ground pressure behavior in roadway-concentrated areas.

(2)Rise-entry barrier coal pillar size

Under the premise of ensuring other conditions remained unchanged, the size of the rise-entry barrier coal pillar was changed. It was set to 150, 190, 230, and 270 m, respectively. The stress and displacement changes of the surrounding rock of the roadway after coal face mining and the influence law of the size of the rise-entry barrier coal pillar on ground pressure behavior were obtained.

#### 3.3.3. Time Factor

To determine the influence of the creep time on the mine pressure behavior, the principle of the only variable was followed. The creep time was set to 30, 60, 90, and 120 days. The stress and displacement changes of roadway-surrounding rock after coal face mining and the influence law of the creep time on the ground pressure behavior were obtained.

## 4. Results

### 4.1. Appearance Law of Ground Pressure Behavior in Roadway-Concentrated Areas under Actual Engineering Conditions

To determine the change of ground pressure behavior in roadway-concentrated areas during the ground pressure behavior of the No. 360801 coal face, the advancing distance of the coal face was set as 100, 200, 300, 400, and 512 m, respectively (the size of the rise-entry barrier coal pillar was 190 m). The roof-to-floor convergence, two-side convergence, maximum concentrated stress, and their positions of the four roadways of the −706-rail crosscut, rail rise-entry, transport rise-entry, and return air rise-entry were monitored and compared. The specific results are shown in Figure 6 and Figure 7.

It can be seen from Figure 6 that with the increase of the advancing distance of the coal face, for the roof-to-floor convergence, each roadway did not change before the advancing distance was less than 300 m. When the advancing distance was greater than 300 m, there was a significant increasing trend, and the two-sided convergence showed a gradually increasing trend from the beginning. In general, when the coal faces advances to 512 m, the −706-rail crosscut, rail rise-entry, transport rise-entry, and return air rise-entry the roof-to-floor convergence increased by 7.66%, 5.11%, 8%, and 4.06%, respectively, compared with 100 m. The two-sided convergence increased by 117.32%, 103.8%, 128.6%, and 59.65%, respectively. Combined with the two above sets of data, it can be seen that with the increase of the advancing distance of the coal face, the influence on the two-sided convergence was greater than that of the roof-to-floor convergence, but the basic trend remained unchanged.

It can be seen from Figure 7 that with the continuous advancement of the coal face, the maximum concentrated stress in the roadway-concentrated areas gradually increased, and the position of the stress concentration was constantly approaching the left side of the roadway. The maximum concentrated stress of the transport rise-entry roadway and its distance from the left side showed the maximum values of the four roadways, and the maximum concentrated stress of the return air rise-entry was the smallest. The closest distance between the maximum concentrated stress position and the left side of the roadway was the −706-rail crosscut, and the position of the maximum concentrated stress did not change significantly with the continuous advancement of the coal face.

In summary, with the advance of the coal face, the roof-to-floor convergence, the two-sided convergence, the maximum concentrated stress, and its location in the roadway-concentrated areas significantly changed. The reason is mainly due to the mining of the coal face and the slip dislocation of the super-thick nappe along the thrust fault caused by mining.

### 4.2. Influence Law of Roof Lithology

According to the simulation scheme in Table 2, the roof-to-floor convergence, two-sided convergence, maximum concentrated stress, and its position of the four roadways of the −706-rail crosscut, rail rise-entry, transport rise-entry, and return air rise-entry in the roadway-concentrated areas after the No. 360801 coal face mining were monitored and compared. The specific results are shown in Figure 8 and Figure 9.

It can be seen from Figure 8 that in the roof-to-floor convergence, affected by the roof lithology, each roadway showed a trend of decreasing first and then increasing. As for the two-sided convergence, the −706-rail crosscut, rail rise-entry, and return air rise-entry were less affected by the lithology of the roof, and only the transport rise-entry showed a decreasing trend. Overall, when the roof lithology was stiff, the −706-rail crosscut, rail rise-entry, transport rise-entry, and return air rise-entry roof-to-floor convergence compared to the lithology was very soft, and decreased by 4.49%, 1.1%, 3.63%, and 0.3%, while the two-sided convergence was reduced by 4.01%, 0.7%, 15.37%, and 2.61%, respectively.

It can be seen from Figure 9 that with the continuous enhancement of roof lithology, for the maximum concentrated stress of the three roadways of the −706-rail crosscut, rail rise-entry, and return air rise-entry, the distance between their position and the left side of the roadway did not change much. The maximum concentrated stress of transport rise-entry showed a decreasing trend, but the distance between the maximum concentrated stress and the left side of the roadway was unchanged when the roof lithology was very soft, soft, and softer. Only when the roof lithology changes from softer to stiff will the distance between the maximum concentrated stress and the left side of the roadway be greatly reduced.

### 4.3. Influence Law of Nappe Thickness

According to the simulation scheme, the roof-to-floor convergence, two-sided convergence, maximum concentrated stress, and its position of the four roadways of the −706-rail crosscut, rail rise-entry, transport rise-entry, and return air rise-entry in the roadway-concentrated areas after mining of the No. 360801 coal face with different thicknesses of nappe were monitored and compared. The specific results are shown in Figure 10 and Figure 11.

It can be seen from Figure 10 that the thickness of the nappe had the same influence on the changing trend of the roof-to-floor convergence and the two-sided convergence in the roadway-concentrated areas under the super-thick nappe, both showing a trend of increasing first and then decreasing, and both changes were significant when the thickness of the nappe was 400 m. In general, when the thickness of the nappe was 550 m, the roof-to-floor convergence of the −706-rail crosscut, rail rise-entry, transport rise-entry, and return air rise-entry was increased by 61.89%, 55.7%, 67.88%, and 52.1%, respectively, compared with the thickness of 100 m, while the two-sided convergence was increased by 138.45%, 107.29%, 180.87%, and 75.05%, respectively.

It can be seen from Figure 11 that the thickness of the nappe had the same influence on the variation trend of the maximum concentrated stress and the distance between the maximum concentrated stress and the left side of the roadway-concentrated areas under the super-thick nappe, which increased first and then decreased. The maximum concentrated stress of the four roadways of the −706-rail crosscut, rail rise-entry, transport rise-entry, and return air rise-entry obviously changed when the thickness of the nappe was 400 m. The distance between the maximum concentrated stress of the three roadways of the rail rise-entry, transport rise-entry, and return air rise-entry with the left side of the roadway obviously changed under the condition where the thickness of the nappe was 250 m, and only the −706-rail crosscut changed when the thickness of the nappe was 400 m.

### 4.4. Influence Law of Mining Height

According to the simulation scheme, the roof-to-floor convergence, two-sided convergence, maximum concentrated stress, and its position of the four roadways of the −706-rail crosscut, rail rise-entry, transport rise-entry, and return air rise-entry in the roadway-concentrated areas after mining of the No. 360801 coal face with different mining heights were monitored and compared. The specific results are shown in Figure 12 and Figure 13.

It can be seen from Figure 12 that the influence of the mining height on the roof-to-floor convergence in the roadway-concentrated areas under the super-thick nappe and the two-sided convergence, except for the −706-rail crosscut roadway, was the same, showing a trend of increasing and then decreasing, and both obviously changed when the mining height was 3 m. The two-sided convergence of the −706-rail crosscut was gradually increasing. In general, when the mining height was 7 m, the roof-to-floor convergence of the −706-rail crosscut, rail rise-entry, transport rise-entry, and return air rise-entry was reduced by 10.15%, 10.09%, 12.97%, and 6.35%, respectively, compared to that when the mining height was 1 m. The two-sided convergence increased by 120.8%, 73.29%, 130.08%, and 46.68%, respectively.

It can be seen from Figure 13 that the continuous increase of mining height, −706-rail crosscut, rail rise-entry, transport rise-entry, and return air rise-entry maximum concentrated stress showed a trend of increasing first and then decreasing, and when the mining height was 3 m, the changes were significant. The distance between the maximum concentrated stress and the left side of the roadway decreased first and then increased, and obviously changed when the mining height was 3 m.

### 4.5. Influence Law of Rise-Entry Barrier Coal Pillar Size

According to the simulation scheme, the roof-to-floor convergence, two-sided convergence, maximum concentrated stress, and its position of the four roadways of the −706-rail crosscut, rail rise-entry, transport rise-entry, and return air rise-entry in the roadway-concentrated areas after mining of the No. 360801 coal face with different sizes of the rise-entry barrier coal pillar were monitored and compared. The specific results are shown in Figure 14 and Figure 15.

It can be seen from Figure 14 that the influence of the size of the rise-entry barrier coal pillar on the roof-to-floor convergence in the roadway-concentrated areas under the super-thick nappe showed a trend of increasing first and then decreasing, while the two-sided convergence showed a trend of gradually decreasing. In general, when the size of the rise-entry barrier coal pillar increased to 270 m, the roof-to-floor convergence of the −706-rail crosscut, rail rise-entry, transport rise-entry, and return air rise-entry decreased by 5.61%, 1.76%, 6.55%, and 1.25%, respectively, compared to that of 150 m, while the two-sided convergence decreased by 61.93%, 46.74%, 63.5%, and 25.9%, respectively.

It can be seen from Figure 15 that with the increase of the size of the rise-entry barrier coal pillar, the maximum concentrated stress of the four roadways showed a decreasing trend, and the decreasing speed suddenly increased when the size of the rise-entry barrier coal pillar was greater than 190 m. The distance between the maximum concentrated stress and the left side of the roadway increased first and then decreased, and obviously changed when the size of the rise-entry barrier coal pillar was 190 m.

### 4.6. Influence Law of Creep Time

According to the simulation scheme, the roof-to-floor convergence, two-sided convergence, maximum concentrated stress, and its position of the four roadways of the −706-rail crosscut, rail rise-entry, transport rise-entry, and return air rise-entry in the roadway-concentrated areas after mining of the No. 360801 coal face with different creep times were monitored and compared. The specific results are shown in Figure 16 and Figure 17.

It can be seen from Figure 16 that the creep time showed a gradual increase in the trend of roof-to-floor convergence in the roadway-concentrated areas under the super-thick nappe. In general, when the creep time was 120 days, the −706-rail crosscut, rail rise-entry, transport rise-entry, and return air rise-entry roof-to-floor convergence compared to the creep time of 30 days increased by 78.21, 74.4, 80.56, and 73.95 times, respectively. The two-sided convergence increased by 164.1, 160.9, 167.1, and 151.1 times, respectively. The reason is that due to the influence of the rock layer relationship, the transport rise-entry was the closest to the coal seam roof, followed by the −706-rail crosscut, so the impact on these two roadways was greater than the rail rise-entry and return air rise-entry.

It can be seen from Figure 17 that the creep time had a serious influence on the maximum concentrated stress in the roadway-concentrated areas under the super-thick nappe, showing a gradual increasing trend, while the change range of the transport rise-entry was larger than that of the other three roadways, and the change range of the return air rise-entry was the smallest. The distance between the maximum concentrated stress and the left side of the roadway was almost unchanged before the creep time of 90 days, and when the creep time exceeded 90 days, there was a slightly obvious increasing trend.

### 4.7. Comparative Analysis of Factors

Based on the above numerical simulation results, it can be seen that for the roof-to-floor convergence in roadway-concentrated areas, with the increase of the thickness of the nappe, the mining height, and the size of the rise-entry barrier coal pillar, they all showed a trend of increasing first and then decreasing. With the increase of roof lithology conditions, the opposite trend was shown, and the creep time increased. For the two-sided convergence and the maximum concentrated stress, it increased first and then decreased with the increase of the thickness of the nappe and the mining height, decreased with the increase of the roof lithology and the size of the rise-entry barrier coal pillar, and showed the opposite trend with the increase of the creep time. For the maximum concentrated stress and the distance from the side, it increased first and then decreased with the increase of the thickness of the nappe and the size of the rise-entry barrier coal pillar. It showed an opposite trend with the increase of the mining height, the increase of the roof lithology, and the increase of the creep time.

By comparing the influence of roof lithology, the thickness of nappe, the mining height, the size of the rise-entry barrier coal pillar, and the creep time on the overall ground pressure behavior in the roadway-concentrated areas, it can be seen that the creep time had the greatest influence on the ground pressure behavior in the roadway-concentrated areas, followed by the thickness of the nappe, the mining height, and the size of the rise-entry barrier coal pillar, and the roof lithology had the least influence on the ground pressure behavior of each roadway. At the same time, by comparing the influence degree of the ground pressure behavior of each roadway in the roadway-concentrated areas, it was found that the above influencing factors had the most significant influence on the transport rise-entry, followed by the −706-rail crosscut, the rail rise-entry, and the return air rise-entry, which was the least affected by the various factors.

## 5. Conclusions

(1)According to the actual geological mining conditions of the No. 36 district in the Xinji No. 1 Mine, the influencing factors of ground pressure behavior in roadway-concentrated areas under super-thick nappe were analyzed from three aspects: geological factors, mining technology factors, and time factors. The main influencing factors were determined as roof lithology, the thickness of the nappe, the mining height, the size of the rise-entry barrier coal pillar, and the creep time.(2)With the advancement of the No. 360801 coal face, the roof-to-floor convergence, the two-sided convergence, and the maximum concentrated stress of each roadway in the roadway-concentrated areas gradually increased, the distance between the position of the maximum concentrated stress and the side gradually decreased, and the deformation of the two-sided convergence was significantly greater than that of the roof-to-floor convergence. Among them, the two-sided convergence and the maximum concentrated stress can reach 365.4 mm and 42.2 MPa.(3)By comparing the influence of various factors on the ground pressure behavior in roadway-concentrated areas, it can be concluded that: the greater the thickness of nappe and the greater the mining height, the longer the creep time, the stronger the mine pressure behavior, the greater the size of the rise-entry barrier coal pillar, and the stronger the roof lithology, and the ground pressure behavior tends to be gentle.(4)By comparing the degree of mine pressure behavior in each roadway under the influence of different factors, it can be found that: The intensity of mine pressure behavior in each roadway was mainly affected by the strata. The closer the roadway was to the coal seam roof, the stronger the mine pressure behavior, while the farther away from the coal seam roof, the gentler the mine pressure behavior. Transport rise-entry was the closest to the coal seam roof, followed by the rail crosscut, so the impact on these two roadways was greater than on the rail rise-entry and the return air rise-entry.

## Figures and Tables

**Figure 1 materials-16-00089-f001:**
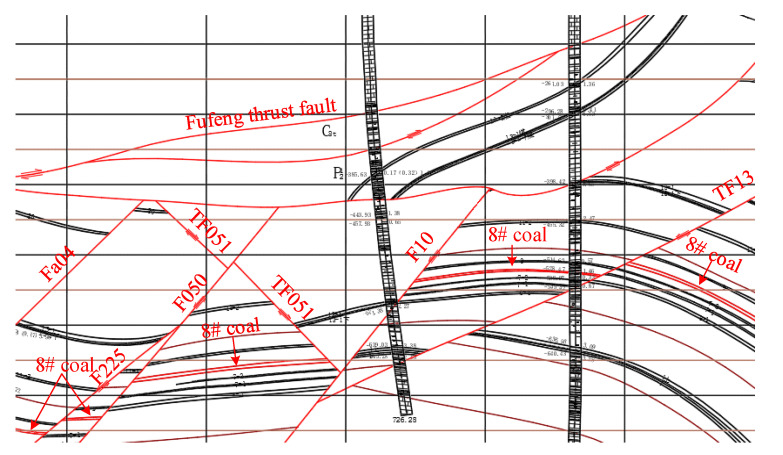
Nappe structure plane diagram.

**Figure 2 materials-16-00089-f002:**
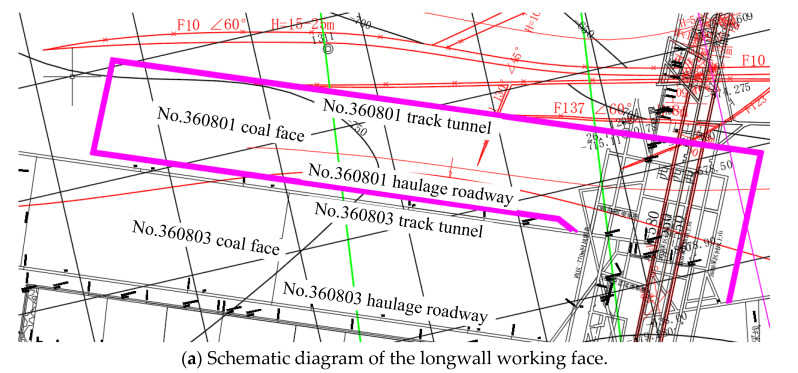
Layout plan of the No. 360801 coal face and roadway-concentrated areas.

**Figure 3 materials-16-00089-f003:**
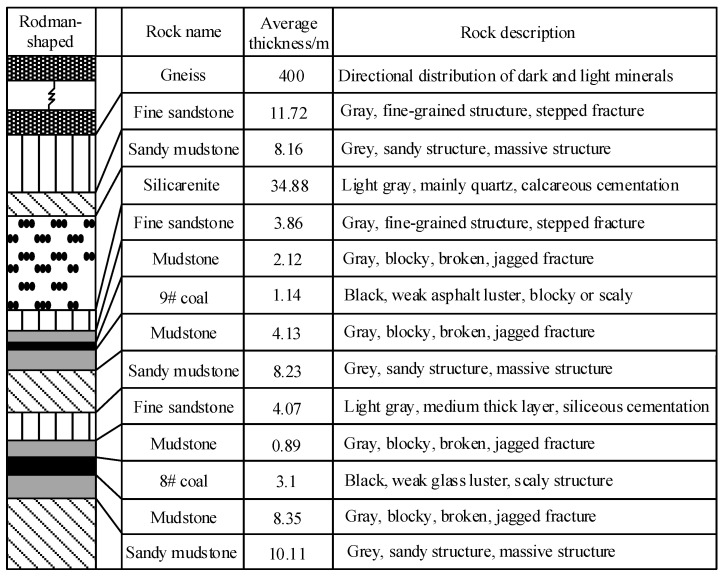
Rock stratum histogram.

**Figure 4 materials-16-00089-f004:**
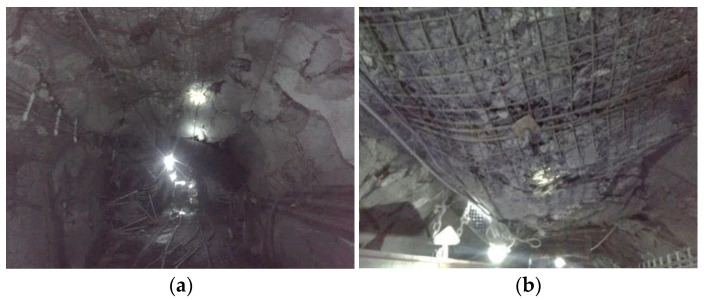
Deformation diagram of partial roadway-surrounding rock. (**a**) Overall damage diagram (**b**) Local damage diagram.

**Figure 5 materials-16-00089-f005:**
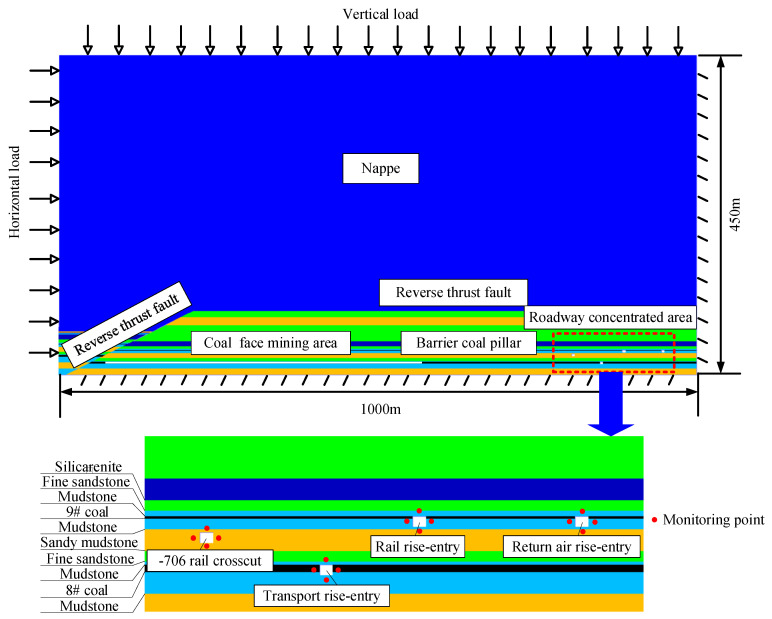
Overall numerical model and roadway-concentrated areas diagram.

**Figure 6 materials-16-00089-f006:**
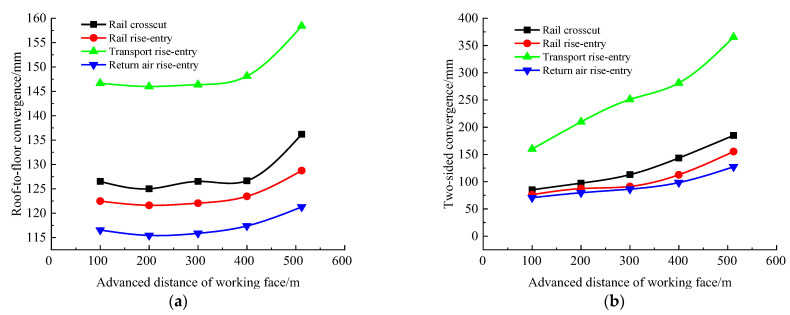
Monitoring results of convergence in roadway-concentrated areas during the advancing process of the No. 360801 coal face. (**a**) Roof-to-floor convergence, (**b**) Two-sided convergence.

**Figure 7 materials-16-00089-f007:**
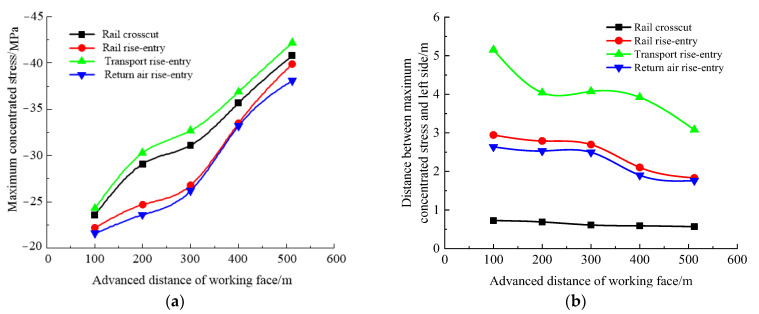
Maximum concentrated stress and its position monitoring results in the advancing process of the No. 360801 coal face. (**a**) Maximum concentrated stress, (**b**) Position of maximum concentrated stress.

**Figure 8 materials-16-00089-f008:**
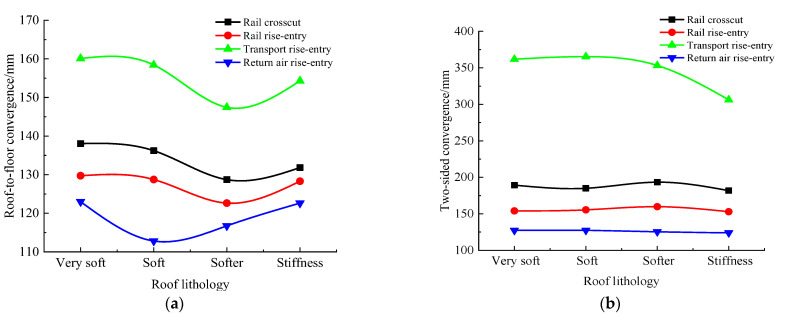
Monitoring results of convergence in roadway-concentrated areas under different roof lithology conditions. (**a**) Roof-to-floor convergence, (**b**) Two-sided convergence.

**Figure 9 materials-16-00089-f009:**
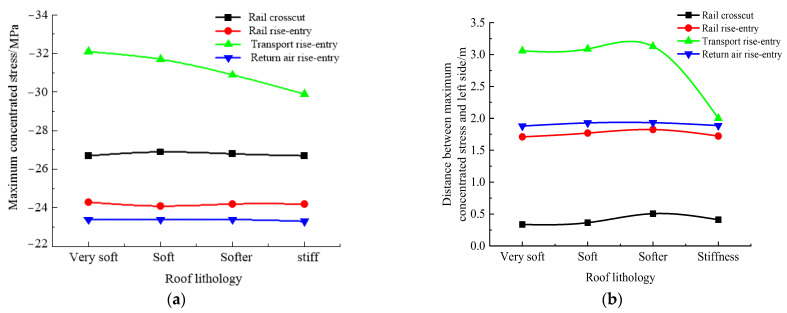
Maximum concentrated stress and its location monitoring results under different roof lithology conditions. (**a**) Maximum concentrated stress, (**b**) Position of maximum concentrated stress.

**Figure 10 materials-16-00089-f010:**
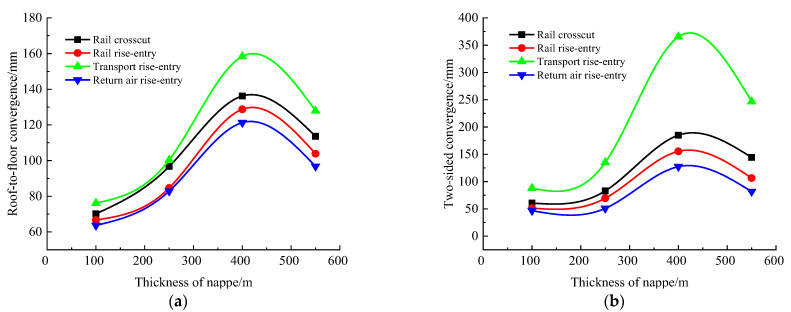
Monitoring results of convergence in roadway-concentrated areas under the different thicknesses of nappe conditions. (**a**) Roof-to-floor convergence, (**b**) Two-sided convergence.

**Figure 11 materials-16-00089-f011:**
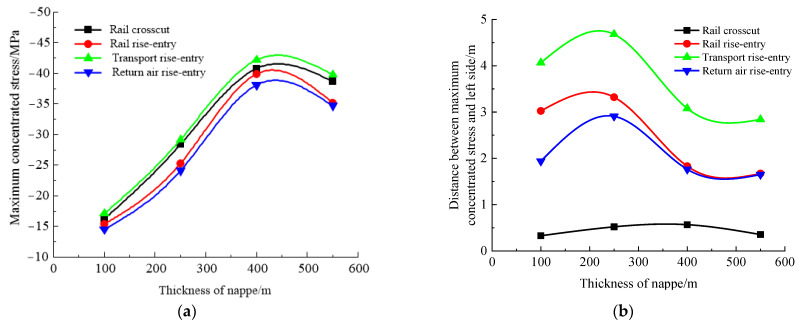
Maximum concentrated stress and its location monitoring results under the different thicknesses of nappe conditions. (**a**) Maximum concentrated stress, (**b**) Position of maximum concentrated stress.

**Figure 12 materials-16-00089-f012:**
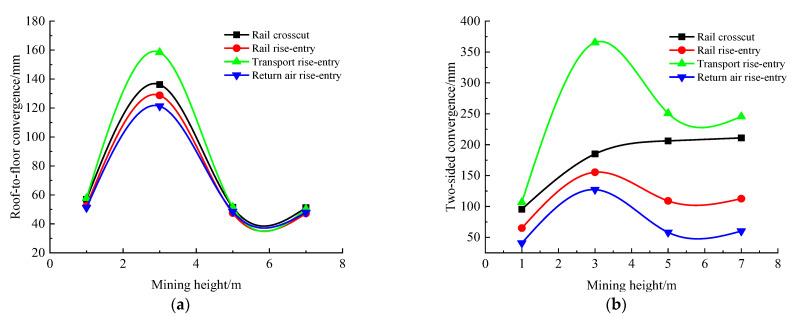
Monitoring results of convergence in roadway-concentrated areas under different mining height conditions. (**a**) Roof-to-floor convergence, (**b**) Two-sided convergence.

**Figure 13 materials-16-00089-f013:**
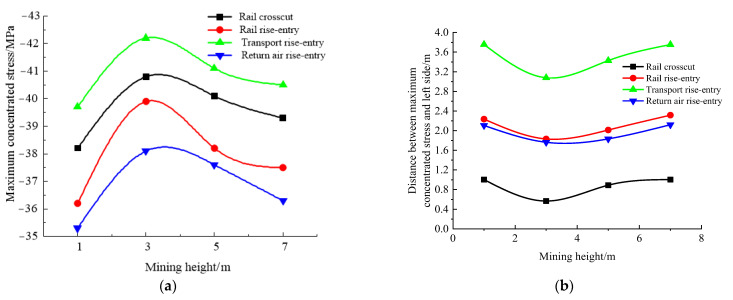
Maximum concentrated stress and its location monitoring results under different mining height conditions. (**a**) Maximum concentrated stress, (**b**) Position of maximum concentrated stress.

**Figure 14 materials-16-00089-f014:**
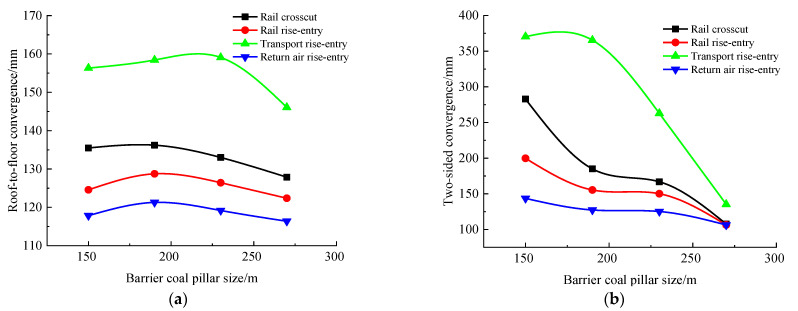
Monitoring results of convergence in roadway-concentrated areas under different sizes of rise-entry barrier coal pillar conditions. (**a**) Roof-to-floor convergence, (**b**) Two-sided convergence.

**Figure 15 materials-16-00089-f015:**
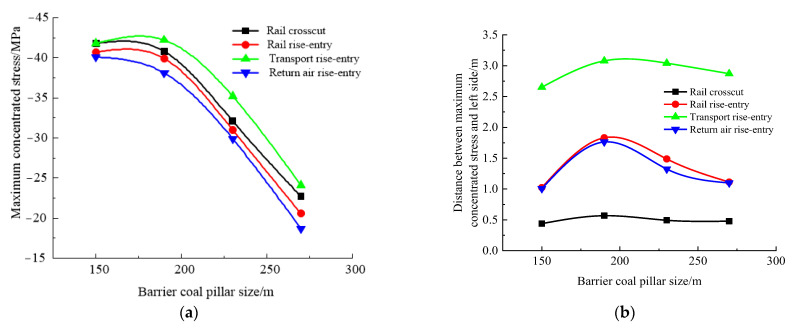
Maximum concentrated stress and its location monitoring results under different sizes of rise-entry barrier coal pillar conditions. (**a**) Maximum concentrated stress, (**b**) Position of maximum concentrated stress.

**Figure 16 materials-16-00089-f016:**
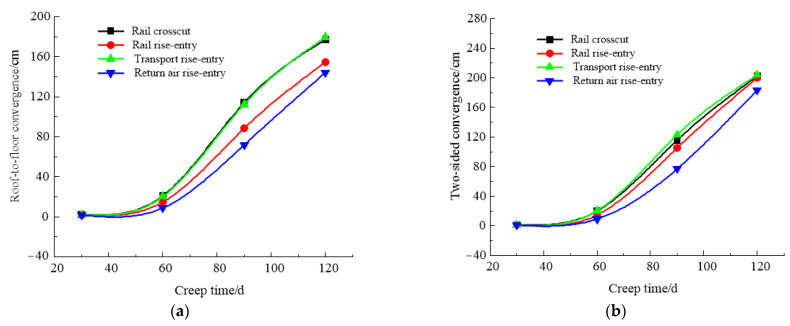
Monitoring results of convergence in roadway-concentrated areas under different creep time conditions. (**a**) Roof-to-floor convergence, (**b**) Two-sided convergence.

**Figure 17 materials-16-00089-f017:**
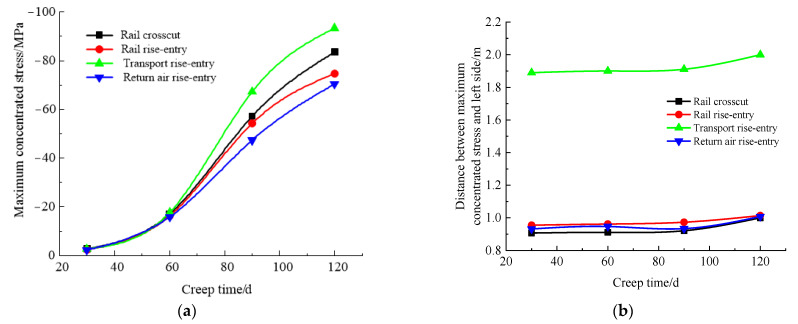
Maximum concentrated stress and its location monitoring results under different creep time conditions. (**a**) Maximum concentrated stress, (**b**) Position of maximum concentrated stress.

**Table 1 materials-16-00089-t001:** Numerical model of rock physical and mechanical parameters table.

	Rock Name	Thickness	Density/kg × m^−3^	Tensile Strength/MPa	Elastic Modulus/GPa	Cohesion/MPa	The Angle of Internal Friction/°	Poisson’d Ratio
1	Gneiss	400	2763	11.2	44.8	15	40	0.25
2	Fine sandstone	12	2532	5.38	37.15	3.2	42	0.27
3	Sandy mudstone	8	2562	2.6	12.08	2.45	40	0.25
4	Silicarenite	35	2423	4.9	21.75	21	75	0.29
5	Fine sandstone	4	2532	5.38	37.15	3.2	42	0.27
6	Mudstone	2	2582	2.0	10.37	1.2	32	0.28
7	No. 9 coal seam	1	1401	0.3	2.79	0.8	29	0.32
8	Mudstone	4	2582	2.0	10.37	1.2	32	0.28
9	Sandy mudstone	8	2562	2.6	12.08	2.45	40	0.25
10	Fine sandstone	4	2532	5.38	37.15	3.2	42	0.27
11	Mudstone	1	2582	2.0	10.37	1.2	32	0.28
12	No. 8 coal seam	3	1378	0.4	1.32	0.8	29	0.31
13	Mudstone	8	2582	2.0	10.37	1.2	32	0.28
14	Sandy mudstone	10	2562	2.6	12.08	2.45	40	0.25

**Table 2 materials-16-00089-t002:** Simulation scheme under different roof lithology conditions.

Name	Simulation Scheme	Specific Parameters					
		Density/kg·m^−3^	Bulk Modulus/GPa	Shear Modulus/GPa	Internal Friction Angle/°	Cohesion/MPa	Tensile Strength/MPa
Roof lithology	Very soft	1378	2.8	1.51	32	0.3	0.945
	Soft	2582	4.23	2.3	40	0.3	2.4
	Softer	2532	8.64	5.69	38	2.1	4.5
	Stiff	2532	12.64	8.69	36	5.2	11.5

## Data Availability

Not applicable.

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
