# Peer review of "Study on Influencing Factors of Ground Pressure Behavior in Roadway-Concentrated Areas under Super-Thick Nappe"

_materials, 2022, doi:10.3390/ma16010089_

Round 1

Reviewer 1 Report

Geological circumstances, mining technical conditions, and time are considered to be the most important determining elements. The principle of ground pressure behaviour in roadway concentrated places, as well as the influence of roof lithology, the thickness of nappe, mining height, size of barrier coal pillar, and creep time, are then all thoroughly studied using the distinct element quantitative mathematical formulation.

1.      Please review lines 24-29 on page no. 1 of the paper to recommend changing the abstract to adopt new facts and desired results.

2.      To the interest of all readers, the authors have given a Nomenclature List.

3.      I suggest adding the paper's structure towards the end of the introduction paragraph. The remaining pages of this manuscript are arranged as follows. Section 2 contains the following: In Section 3, .......... in Section 4, etc.)

4.      The problem statement must also be mentioned.

5.      It seems to be a case study; how should this be researched?  Check pages 2 -4.

6.      provide a brief statement that describes the main factors.

7.      Please check page 8 lines 212-218, and how can that condition be measured?

8.      which point will have the biggest effect on the site, please?  Page 11, figure 9.

9.      The inadequacies of the study must be reiterated, with an emphasis on the conclusions.

10.  The conclusion should include additional details

More comments can be found in the attachment.

Regards

Reviewer 2 Report

 The present manuscript describes the main factors affecting the ground pressure behavior in the roadway concentrated areas under the super-thick nappe. Also discrete element numerical simulation software, a numerical simulation model under different geological and mining conditions is established. A manuscript has a practical application and also provides important theoretical for the next studies. The paper can be accepted for publication after providing the corrections mentioned below.

Point 1. Authors should rephrase the second sentence of the abstract “Taking the no.360801 coal face of Xinji No.1 Mine as the engineering back-12 ground, first, the main influencing factors are determined from the aspects of geological conditions, 13 mining technical conditions and time”. Do not indicate “no.360801 coal face of Xinji No.1”.

Point 2. The same situation is about “the -706-rail” in the Abstract.

Point 3. In the end of the Introduction section authors indicate “The research results are of great significance for guiding the safe mining of 79 no.360801 coal face”. So, the question is if it is the case study? If yes, then indicate it in the paper title.

Point 4. Line 76. You should provide full meaning of the UDEC.

Point 5. The Introduction section seems to be too short. An enhanced literature review is required. Most used literature presented in the Introduction comes from China. What about other experience?

Point 6. Regarding Point 6, please consider the suggested research of authors from Kazakhstan and Ukraine in your paper when enhancing the literature review. I believe they are worth considering in your paper.

Begalinov, A., Almenov, T., Zhanakova, R., & Bektur, B. (2020). Analysis of the stress deformed state of rocks around the haulage roadway of the Beskempir field (Kazakhstan). Mining of Mineral Deposits, 14(3), 28-36. https://doi.org/10.33271/mining14.03.028

Important Issue: The studies of physical and mechanical characteristics and spread of the stress deformed state with due account for the fault around the mine working lead to the following conclusion. It was established that 41.6% of the mine working with due account for the fault zone is unstable, and 58% of it is a more stable part. This means that stress state changes drastically throughout the mine development. The largest part of the mine working is reliable, although according to the project it is completely supported (100%) in compliance with category III (monolithic concrete support). A certain stress deformed state ensures the adjustment of the support’s length with due account for specific data. In line with these values it is recommended to use the specific design of support with adjustable resistance, where only 23% of the mine working length corresponds to support category III. Thanks to the use of such supports, it becomes possible to control the rock pressure in the complex mining geological conditions of the Beskempir field.

Babets, D., Sdvyzhkova, O., Shashenko, O., Kravchenko, K., & Cabana, E.C. (2019). Implementation of probabilistic approach to rock mass strength estimation while excavating through fault zones. Mining of Mineral Deposits, 13(4), 72-83. https://doi.org/10.33271/mining13.04.072

The paper addresses the rock mass state estimation while excavating a cross-heading through the area of regional fault “Bohdanivskyi” based on probabilistic approach to assessing the rock strength.

Point 7. Please loo at the figure 2 b and c. Especially “coal face”. Truth be told it is not coal face but longwall (e.a. panel not line).

Point 8. On the figure 4 is shown two figures, so indicate what is shown on the figure (a) and (b).

Point 9. Subsection 3.2.3. Time factor – “Creep time is set to 30 d, 60 d, 90 d and 120 d”. Why such time was set up?

Point 10. The content of the manuscript is similar to that of a case study. The knowledge contained here may be useful for engineers, students, and scientists, searching for any knowledge related to mining engineering, which is the most important value of the manuscript. In general, I must admit that a very good study was performed, and I will recommend your paper for publication after careful revision.

Reviewer 3 Report

Comments for Authors

1) In the header and footer of the article, the year is incorrect ® 2021,

2) Edit the text and structure of the article according to the instructions of MDPI Materials :

                                                           - shorten the abstract, maximum 200 words,

                                                   - correct the structure of the article to:

                                                                                1. Introduction

                                                                            2. Materials and Methods

                                                                            3. Results

                                                                            4. Discusions

                                                                            5. Conclusions

                                    - references – Page 17, References, correct the font style according to the requirements of the MDPI Materials magazine.

3) Page 2, from line 82 - insert a small map of the research area,

4) Page 3,4,9,10,11,13,12,14 and 15, change everywhere ® for example from Fig.1 to Figure 1.

5) Page 7 and 8, table 1 and 2, unify Density kg×m-3 ® kg/m3,

6) Page 8, table 2, edit the font (size, thickness) as in table 1,

7) In the Conclusion, briefly add the science of your research, benefits, advantages - disadvantages, practical application. Perspectives in the continuation of research and trials.

Round 2

Reviewer 2 Report

Dear authors, I am more than satisfied with the corrections provided by you.

This study is an important contribution to sustainable mining. Congratulations to the authors.